# Assessment on Temporal and Spatial Variation Analysis of Extreme Temperature Indices: A Case Study of the Yangtze River Basin

**DOI:** 10.3390/ijerph182010936

**Published:** 2021-10-18

**Authors:** Guangxun Shi, Peng Ye

**Affiliations:** 1School of Geography, Nanjing Normal University, Nanjing 210023, China; 161301021@njnu.edu.cn; 2Key Laboratory of Virtual Geographic Environment, Ministry of Education, Nanjing Normal University, Nanjing 210023, China; 3Jiangsu Center for Collaborative Innovation in Geographical Information Resource Development and Application, Nanjing 210023, China; 4State Key Laboratory Cultivation Base of Geographical Environment Evolution (Jiangsu Province), Nanjing 210023, China; 5Urban Planning and Development Institute, Yangzhou University, Yangzhou 225127, China; 6College of Civil Science and Engineering, Yangzhou University, Yangzhou 225127, China

**Keywords:** extreme temperature indices, spatial heterogeneity, abrupt, prediction, disaster risk, Yangtze River Basin

## Abstract

Extreme temperature change is one of the most urgent challenges facing our society. In recent years, extreme temperature has exerted a considerable influence on society and the global ecosystem. The Yangtze River Basin is not only an important growth belt of China’s social and economic development, but also the main commodity grain base in China. The purpose of this study is to study the extreme temperature indices in the Yangtze River Basin. In this study, the Mann–Kendall nonparametric test and R/S analysis method are used to analyze the spatial and temporal variation characteristics of major extreme temperature indices in the Yangtze River Basin from 1970 to 2014. The main conclusions are drawn as follows: (1) The occurrence of cold days (TX10), cold nights (TN10), ice days (ID), and frost days (FD) decrease at a rate of −0.66–−2.5 d/10a, respectively, while the occurrence of warm days (TX90), warm nights (TN90), summer days (SU), and tropical nights (TR) show statistically significant increasing trends at a rate of 2.2–4.73 d/10a. (2) The trends of the coldest day (TXn), coldest night (TNn), warmest day (TXx), warmest night (TNx), and diurnal temperature range (DTR), range from −0.003 to 0.5 °C/10a. (3) Spatially, the main cold indices and warm indices increase and decrease the most in the upper and lower reaches of the Yangtze River Basin. (4) DTR and TN90 show no abrupt changes; the main cold indices changed abruptly in the 1980s and the main warm indices changed abruptly in the late 1990s and early 2000s. (5) The extreme temperature indices are affected by the atmospheric circulation and urban heat island effect in the Yangtze River Basin. Relative indices and absolute indices will continue to maintain the present trend in the future. In short, the main cold indices of extreme temperature indices show a decreasing trend, the main warm indices of extreme temperature indices show an increasing trend, and cold indices and warm indices will continue to maintain the present trend in the future in the Yangtze River Basin. Extreme temperature has an important impact on agriculture, social, and economic development. Therefore, extreme temperature prediction and monitoring must be strengthened to reduce losses caused by extreme temperature disasters and to promote the sustainable development in Yangtze River Basin.

## 1. Introduction

According to the fifth assessment report of the Intergovernmental Panel on Climate Change, from 1880 to 2012, the global mean surface temperature increased significantly, with an increase of 0.85 °C. The annual mean temperature from 2003 to 2012 increased by 0.78 °C compared with that from 1850 to 1900 [1]. By the end of the century, global mean surface temperature may increase, and it is possible that the surface temperature will rise by a maximum of 2 °C [2]. The rise in temperature has become an indisputable fact. Global warming may increase the frequency and intensity of extreme weather and climate events [3]. Compared with the increase in mean temperature, regional extreme temperature changes have a more significant and direct impact on society and ecosystems.

At present, extreme temperature events have become commonplace on a global scale. Many scholars have analyzed the extreme temperatures in different regions of the world and they found that Australia’s extremely cold days and frost days are on a downward trend, and extreme high temperature days are on the rise [4]. Some scholars have argued that mean annual temperature over southern Canada has increased by an average of 0.98 °C, with the largest warming during winter and early spring, since 1900 [5]. Other studies suggest that the number of warm days has increased, and the number of cold nights has decreased significantly, in Uruguay [6]. Some scholars believe that number of frost days in Europe shows a decreasing trend, and the number of summer days shows an increasing trend [7]. The number of cold days and cold nights in Morocco is decreasing, while the number of warm days and warm nights is increasing [8]. Similar work has found the percentage of warm nights/days in 70% of stations in South and Central Asia increased significantly, while the percentage of cold nights/days is decreased significantly [9]. Meanwhile, Chinese scholars have also carried out research on extreme temperatures: the rising tendency of the minimum temperature in Tianjin in winter and spring is the most obvious; the longest cold nights and frost days in winter and spring are significantly reduced, and the number of long cold nights is significantly reduced [10]. On a small regional scale, it was observed that the increased magnitude of extreme high temperatures in Beijing is significantly less than that of extreme low temperatures, and the increase in the diurnal indices is less than the night indices of extreme temperatures [11]. On a larger regional scale, it was observed that the indices of cold nights (days) in Northeast China, Loess Plateau, Xinjiang region, Pearl River Basin and Yellow River Basin showed a downward trend, and the indices of warm nights (days) showed an increasing trend, but their increased magnitudes and spatial distributions were varied [12,13,14,15]. In addition, some scholars believe that the extreme minimum temperature, annual mean temperature, and extreme maximum temperature in China’s mainland have significantly increased [16,17]. All the above studies demonstrate that in the global warming environment, the occurrence of extreme high temperatures increases, while that of the cold indices decreases. However, there were varieties in the increasing magnitudes of extreme temperatures and their spatial patterns. Therefore, this research on regional extreme temperatures is of practical significance. The Yangtze River Basin is the largest basin in China, and it is an important economic belt, spanning the three major economic zones of southwest, central and eastern China [18]. Therefore, the study of extreme temperature in the Yangtze River Basin is very important.

However, in the past few decades, with the rapid economic development, population increase, urban expansion, and changes of land use/land cover, the climate conditions in the Yangtze River Basin have changed [19,20]. According to the previous studies, the average temperature in the Yangtze River Basin is increased significantly [21]. The extreme temperatures of the Yangtze River Basin in winter and summer show an upward tendency [22]. Some scholars analyzed the daily minimum temperature, daily average temperature, and daily maximum temperature in the Yangtze River Basin, and found indicators on an obvious increasing tendency in the late 1980s [23]. Other scholars have argued that the number of warm days and warm nights were increasing, while the number of cold days and cold nights were decreasing [24]. Moreover, other studies suggest extreme temperature indices of the Yangtze River Basin and concluded that the warming range of the cold indices was greater than that of the warm indices, and the warming range of the night indices was greater than that of the diurnal indices [25]. However, it is found that there is a strong correlation between altitude and the changing trend of extreme temperature. Above 350 m, the warming rate increases with the increase of altitude, while below 350 m, the warming rate decreases with the increase of altitude [26]. In terms of spatial pattern, the meteorological stations with obvious trends of extreme temperature indices were mostly in Sichuan Basin and the lower Yangtze River Basin, and almost all the extreme temperatures showed the maximum tendency in spring and winter [27]. In addition, many scholars have conducted research on sub basins or provincial administrative units in the Yangtze River Basin [28,29,30]. The above studies on the extreme temperature in the Yangtze River Basin mainly focus on (1) the average temperature, the minimum temperature, and the maximum temperature in the Yangtze River Basin; (2) the temporal and spatial variation of extreme temperature indices in the Yangtze River Basin, or the sub basins or provincial administrative units in the Yangtze River Basin. There is a lack of prediction of the future extreme temperature in the Yangtze River Basin.

Another reason to study extreme temperatures in the Yangtze Basin is that the frequent occurrence of heat waves has become an important public health problem [31]. The IPCC assessment report points out that since the mid-20th century, due to global warming and urban heat island effect, summer heat waves have occurred frequently in the world [1]. Studies show that heat waves are linked to heatstroke and heat-related illnesses, leading to an increase in deaths among residents. There are many studies on the health effects of high temperature heat waves in Europe and North America. During the heat wave, a large number of people die directly or indirectly. These deaths are mainly concentrated in the elderly and in persons with underlying diseases [32,33,34]. In China, studies on the health effects of heat waves have focused on large cities, such as Wuhan City, Nanjing City, and Shanghai City in the Yangtze River basin [35,36,37]. The Yangtze River basin is an important economic zone and urban zone in China. The study of extreme temperature can help to prevent and monitor the impact of heat wave on public health.

Therefore, this paper selected 131 meteorological stations in the Yangtze River Basin from 1970 to 2014 for comprehensive analyses. The main purposes of this study are (a) to know the time when there are abrupt changes of the main extreme temperature indices in the Yangtze River Basin, (b) to select abnormal extreme temperature indices by using Rescaled range analysis (R/S) and box-plot method to predict the main extreme temperatures, (c) to discuss the causes of extreme temperature in the Yangtze River Basin. (d) The study carries theoretical value for extreme temperatures research. It is also conducive to a better understanding of extreme temperatures and influencing factors in various regions of the Yangtze River Basin. In addition, the empirical analysis provides the basis for the government to formulate corresponding policies, to reduce losses caused by extreme temperature disasters, and to promote sustainable development in the Yangtze River Basin.

## 2. Study Area, Data, and Methods

### 2.1. Study Area

The Yangtze River Basin (90°33′~122°25′ E, 24°30′~35°45′ N) (Figure 1) refers to the vast area through which the main and tributaries of the Yangtze River pass. It spans 19 provincial administrative units in three economic zones of China (the eastern economic zone, central economic zone and western economic zone). It is the third largest basin in the world with a total area of 1.8 million square kilometers, which accounts for 18.8% of China’s land area. The terrain of the basin is fluctuant, high in the west and low in the east, showing a three-step shape. Except for some high-altitude areas such as the western Sichuan Plateau and the headwaters of the Yangtze River, most of the basin belongs to the middle and north subtropical monsoon climate. The Yangtze River Basin is the main driving axis of China’s economic development. The Yangtze River economic zone, together with the eastern coastal areas, has become the main axis of China’s inverted “T” shape economy and plays an extremely important strategic role in China’s social and economic development [38]. The Yangtze River Basin is also a densely populated area in China. Strengthening the research on extreme temperature is conducive to reducing the impact of extreme temperature on public health problems. Moreover, the Chengdu Plain in the Mintuo River Basin and the Hanjiang Plain in the Jianghan River Basin, etc., in the Yangtze River Basin, are important commodity grain bases, which play an important role in China’s food security. In short, studying extreme temperature in the Yangtze River Basin is conducive to ensure the sustainable development of society and economy.

### 2.2. Data

There are 164 national meteorological stations in the Yangtze River Basin. The resource of daily observations of the stations utilized in this study were from the China Meteorological Data Service Center (CMDC). Considering the start and ending time of meteorological data records at the stations and that some stations have been demolished or relocated, this study selected real data from 131 stations in the basin from 1970 to 2014 so as to ensure the integrity and consistency of meteorological data (Figure 1). For a few stations with missing data less than 7 days within a month, the regression method, based on the adjacent stations, was used to interpolate [39]. The basin boundary data are extracted mainly based on the DEM data of 1 km resolution.

DMSP (Defense Meteorological Satellite Program)/OLS (Operational Linescan System) night light data were obtained from the NOAA National Geophysical Data Center. DMSP/OLS data are cloudless, non-radiation-calibrated night light images, including three kinds of annual average images: cloudless observation frequency, average light, and stable light. The spatial resolution of DMSP/OLS noctilucent data is 30″ (arc second, about 1 km), and the pixel value is distributed in the interval (0, 63). It contains persistent light sources such as cities and towns, and excludes the influence of accidental noise such as moonlight, fire, and oil and gas combustion. DMSP/OLS data reflect the nighttime power consumption of public infrastructure, commerce, and residents. Therefore, it is closely related to the intensity of human economic activities and can reflect the development of urbanization, population, and industry. The DMSP/OLS data analyzed in this paper included F10 1992 and F18 2013 collected by two satellite (F10 and F18) images.

### 2.3. Methods

#### 2.3.1. Extreme Temperature Indices

The definitions of extreme temperature indices are based on the detection and indicators of climate change defined by the World Meteorological Organization’s Climatology Committee (WMO-CC), the World Climate Research Program (WCRP), Climate Variability and Predictability Program (CLIVAR), and Expert Team for Climate Change Detection Monitoring and Indices (ETCCDMI). This method has been widely used in extreme climate events research [40,41]. Thirteen extreme temperature indices were selected in this paper (Table 1).

#### 2.3.2. Mann–Kendall (M–K) Trend and Abrupt Changes Analysis

The nonparametric Mann–Kendall (M–K) test method is used here for trend analysis [42]. At present, the M–K test method is mainly used for analyses of abrupt changes in temperature when the significance level is *p* < 0.05. In the process of the M–K abrupt change test, if the positive and negative series have several obvious intersections in the confidence interval, by combining the moving *t*-test, the intersection can be determined to be a real abrupt change point [43]. The following is the introduction to the Mann–Kendall trend analysis method:

Provided that $x_{1},x_{2},\cdots ,x_{n}$ are the data values in time series and n is the number of data points, in the Equation
(1)$$S=\overset{n-1}{\underset{k=1}{\sum }}\overset{n}{\underset{j=k+1}{\sum }}\mathrm{sgn}(x_{j}-x_{k})$$

$x_{j},x_{k}$ are the measured values of *j* and *k*, and *k* > *j*. And,
(2)$$\mathrm{sgn}(x_{j}-x_{k})=\{\begin{array}{ccc} 1 & , & x_{j}-x_{k}>0\\ 0 & , & x_{j}-x_{k}=0\\ -1 & , & x_{j}-x_{k}<0 \end{array}$$

Then,
(3)$$Var\left(S\right)=\frac{n\left(n-1\right)\left(2n+5\right)-\sum_{k=1}^{n}k\left(k-1\right)\left(2k+5\right)}{18}$$
(4)$$Z=\{\begin{array}{c} \frac{S+1}{\sqrt{Var\left(S\right)}},S<0\\ 0{}^{\circ}{}^{\circ}{}^{\circ},S=0\\ \frac{S-1}{\sqrt{Var\left(S\right)}},S>0 \end{array}$$
where *Z* is the standard normal test statistic and positive values of *Z* indicate increasing trends while negative *Z* values show decreasing trends. In this study, the specific significance level is *p* < 0.05, which means *Z* > 1.96 or *Z* < −1.96.

When *Z* does not equal zero, Sen’s slope estimator is used to define the trends. The formula is:(5)f(t)=Qt+B

The value of *Q* indicates the steepness of the trend, *B* is a constant, and *t* is a data year.
(6)$$Q=Median\frac{x_{j}-x_{k}}{j-k}$$

#### 2.3.3. R/S Analysis

Rescaled range analysis (R/S) is a classification-structured analysis method for processing time series [44]. Studies have shown that natural phenomena such as precipitation, temperature, and sunspots all have a Hurst effect. H is the Hurst exponent. When 0.5 < H < 1, it describes a dynamically persistent, or trend reinforcing series; the greater the H value is, the stronger and more persistent. When H = 0.5, it means that the time series is an independent random process, which indicates that the current trend will not affect the future trend. When 0 < H < 0.5, it describes an anti-persistent, or a mean reverting system; the smaller the H value, the stronger the anti-persistent [45].

#### 2.3.4. The Sliding *t*-Test

The sliding *t*-test detects mutations by examining whether there is a significant difference between the mean values of two groups of samples. For a time series *x* with *n* sample sizes, to set an artificial time as a reference point, the samples of two sequences *x*_1_ and *x*_2_ before and after the reference point, and are *n_1_* and *n_2_*; the mean values of the two sequences are *x*_1_ and *x*_2_ and the variances are $s_{1}^{2}$ and $S_{2}^{2}$, respectively. Define statistics:(7)$$t=\frac{\bar{x_{1}}-\bar{x_{2}}}{s\cdot \sqrt{\frac{1}{n_{1}}+\frac{1}{n_{2}}}}$$
of which
(8)$$s=\sqrt{\frac{n_{1}s_{1}^{2}+n_{2}s_{2}^{2}}{n_{1}+n_{2}-2}}$$
where the equation follows the *t*-distribution of freedom *V* = *n*_1_ + *n*_2_ − 2.

In order to avoid the drift of mutation points caused by arbitrary selection of subsequence length, the length of the subsequence can be changed repeatedly for experimental comparison when using this method to improve the accuracy of the calculation results.

## 3. Results

### 3.1. Temporal and Spatial Variations of Extreme Temperatures

#### 3.1.1. Temporal Variation of Relative Indices

From the time scale representation, the relative indices of the Yangtze River Basin changed significantly from 1970 to 2014 (Figure 2). Cold days (TX10) and cold nights (TN10) were in a significant downward trend, and the trends were −2.2 d/10a (*p* < 0.05) and −3.6 d/10a (*p* < 0.001), respectively, tested by the significance level. The number of cold days reached its minimum in 1999, at about 21.7 d, and then increased in fluctuation. Warm days (TX90) and warm nights (TN90) obviously increased, and the trends were 4.73 d/10a (*p* < 0.001) and 3.81 d/10a (*p* < 0.001), respectively, among which the warm days (TX90) showed a significant upward trend after 2003. Besides, the number of warm nights reached a minimum in 1993, at about 21.8 d, and then rose in volatility.

#### 3.1.2. Spatial Variation of Relative Indices

Spatially, cold days (Tx10) of more than 98% of the stations showed decreasing tendencies, of which 35.8% of the stations passed the significance level test (*p* < 0.05). The decreased magnitude of cold days in the Jinsha River basin was the most apparent, with the trends above −4 d/10a, followed by the middle reaches of the mainstream. More than 97% of stations had a decreasing trend in cold nights (TN10), with 74% of them passing the significance level test (*p* < 0.05). From the perspective of the whole river basin, there were particularly significant decreasing tendencies in the Jinsha River Basin and the middle reaches of the main stream area, and the Hanshui River Basin, with the trends exceeding −4.5 d/10a. However, the increasing magnitudes of extreme temperatures and their spatial patterns vary. The remaining 98.5% of the stations showed increasing tendencies, and 72.5% of the stations passed the significance level test (*p* < 0.05), which mainly concentrated in the Jinsha River Basin, the Taihu Lake Basin, and the southern stream of the Minjiang River and the Tuojiang River. In addition, warm nights (TN90) of 96.9% of the stations showed increasing tendencies, all of which passed the significance level test, and their change magnitudes were generally between 3 d/10a–6 d/10a. The change magnitudes of the regions in the basin were significant except for the slight increased magnitudes in the southern and upper mainstream of the Jialing River Basin (Figure 3).

### 3.2. Temporal and Spatial Variations of Absolute Indices

#### 3.2.1. Temporal Variation of Absolute Indices

The number of ice days (ID) showed a slight decreasing trend fluctuation, and the trend was −0.66/10a (*p* < 0.001). The number of frost days (FD) presented a fluctuating decreasing tendency with a trend of −2.5 d/10a (*p* < 0.001), the lowest value was 41.2 d in 1991, followed by a significant fluctuating decrease. Summer days (SU) showed a fluctuating upward trend with a trend of 2.2 d/10a (*p* < 0.001) reaching the minimum 142.5 d in 1982, followed by an increase in volatility. Tropical nights (TR) showed a fluctuating increasing trend with a trend of 2.8 d/10a (*p* < 0.001), reaching its lowest value 100.1 d in 1976 and then showing a significant increasing tendency in fluctuation (Figure 4).

#### 3.2.2. Spatial Variation of Absolute Indices

From the regional scale representation, there were 19 stations with no changing trend in the number of ice days (ID), and there were no freezing days, which mainly distributed in the southern part of the Jinsha River Basin, the southern part of the Min-tuojiang River Basin, and most of the Jialing River Basin (Figure 5). A total of 76.3% of the stations showed a downward trend in ice days (ID), 32% of which passed the significance level test (*p* < 0.05), and the change magnitudes ranged from 0–0.5 d/10a. Stations in the Dongting Lake basin and its middle reaches of the mainstream showed an increasing trend, but it did not pass the significance level (*p* < 0.05). However, the stations that passed the test were mainly distributed in the Taihu Basin and the Poyang Lake Basin. The number of frost days (FD) decreased significantly in the whole basin and 96.9% of the stations showed a downward trend, among which 73.3% of the stations passed the significant level. The significant decreased magnitudes were mainly distributed in the Jinsha River Basin, Hanshui River Basin, Taihu Lake Basin, and the middle reaches, with the change magnitudes between −3–−7 d/10a. There were six stations in the basin with no change (the trend = 0) in summer days (SU), which mainly distributed in the high-altitude area of Qinghai Plateau. The summer days in the Taihu Lake Basin and most of the Poyang Lake Basin showed a decreasing trend with a trend of 0–−2.5 d/10a. Most of the rest of the region showed an increasing tendency, and they were distributed in the central part of the Yangtze River Basin, including Dongting Lake Basin, Hanshui River Basin, Jialing River Basin, Wujiang River Basin, Mintuojiang River Basin, and most of the southern part of the Jinsha River Basin.

There were 21 stations showing no changes (the trend = 0) in terms of tropical nights (TR), and were mainly distributed in the Jinsha River basin. Tropical nights (TR) in other areas in the basin generally showed an increasing trend, which were mainly distributed in the middle and lower reaches of the Poyang Lake Basin, the Hanshui River Basin, the Middle and the Lower Mainstream Area, the Taihu Lake Basin, and the Wujiang River Basin, with a change magnitude of 2 d/10a–6 d/10a and 84.6% of the stations having passed the significance level test.

### 3.3. Temporal and Spatial Variations of the Extremal Indices

#### 3.3.1. Temporal Variation of the Extremal Indices

The minimum value of daily maximum temperature (TXn) increased significantly with a trend of 0.39 °C/10a (*p* < 0.001), reached the lowest value of −3.5 °C in 1977 and increased in fluctuating tendencies after 1977 (Figure 6). The minimum value of daily minimum temperature (TNn) rose obviously with a trend of 0.5 °C/10a, and also showed the minimum in 1977, which was −9.1 °C, and after 1977 it increased in volatility. The maximum value of daily maximum temperature (TXx) evidently increased with a trend of 0.27 °C/10a (*p* < 0.001), which was extremely low in 1993 with a minimum of 34.7 °C, and increased in fluctuating tendencies after 1993. The maximum value of daily minimum temperature (TNx) showed a significant increasing trend, which was 0.24 °C/10a. In 1974, it reached its lowest value of 24.8 °C, and increased in volatility after that year. There were slight decreasing tendencies in diurnal temperature range (DTR) and the trend was −0.003 °C/10a (*p* = 0.51). In 1989, its value achieved the minimum of 6.48 °C and after that year it increased in volatility.

#### 3.3.2. Spatial Variation of Extremal Indices

The minimum value of daily maximum temperature (TXn) of 128 stations in the basin had increasing tendencies and 61% of the stations passed the significance level test (Figure 7). In particular, the increase was significant in most of the Dongting Lake Basin and the main stream of the middle reaches of Taihu Lake Basin, with a change magnitude of 0.4–0.6 °C/10a. The minimum value of the daily minimum temperature (TNn) of 95.4% of the stations showed an increasing trend, with 44% of the stations having passed the significance level test (*p* < 0.05). The areas with large increase of TNn were mainly distributed in the southern part of the Jinsha River Basin, the Hanshui River Basin, the middle reaches of the mainstream, the Poyang Lake Basin, and the Taihu Lake Basin, with a change magnitude of 0.2–0.6 °C/10a. Among them, the Dongting Lake Basin had a relatively slight change magnitude, which was between 0–0.2 d/10a. The maximum value of daily maximum temperature (TXx) was increased in all stations except for three stations, and 45% of the stations passed the significance level test (*p* < 0.05).

The areas with a large increase of TXx were mainly concentrated in the Jinsha River Basin, Minjiang River Basin, and Taihu Lake Basin, with an increased magnitude of 0.6–0.8 °C/10a. The maximum value of the daily minimum temperature (TNx) of 90% of the sites in the basin showed increasing tendencies, of which 73.5% of the stations passed the significance level test (*p* < 0.05) with an increased magnitude of 0.2–0.6 °C/10a. The areas with a larger increase of TNx were mainly distributed in the middle stream, the downstream stream, the Poyang Lake basin, the Hanshui River Basin, and the Taihu Lake Basin; diurnal temperature range (DTR) of 46.8% of the stations showed an increasing trend, and 33% of the stations were mainly distributed in the Jinsha River Basin, Wujiang River Basin, and Hanshui River Basin through the significance level test (*p* < 0.05), with a change magnitude of more than 0.15 °C/10a. DTR of 53.2% of the stations showed a downward tendency, of which 44.9% of the stations passed the significance level test (*p* < 0.05) and they were mainly concentrated in the Mintuojiang River Basin and the middle reaches of the mainstream, with a change magnitude of 0.15 °C–0.3 °C/10a.

### 3.4. Analysis of Abrupt Changes of Extreme Temperature Indices

#### 3.4.1. Relative Indices

In this paper, the M–K test analysis is carried out on four relative indices in the Yangtze River Basin (Figure 8). There are two intersection points of the positive and negative serial curves of the cold days (TX10) within the confidence line, and a sliding *t*-test is performed in this paper, none of which show abrupt change points; positive and negative serial curves of other relative indices have only one intersection within the confidence line. Cold nights (TN10) has a crossing point in the confidence interval, and there was an abrupt change in 1987. There is an intersection point of warm days (TX90) in the confidence interval, and an abrupt change occurred in 2003. The warm nights (TN90) has an intersection out of the confidence interval and this point is not an abrupt change point.

#### 3.4.2. Absolute Indices

In this paper, the M–K test analysis is carried out on the four absolute indices in the Yangtze River Basin (Figure 9). There is one intersection point in the positive and negative serial curves of ice days (ID) within the confidence line, and there was an abrupt change in 1992. The sliding *t*-test is performed in this paper, and the abrupt change point appeared in 1992. The positive and negative serial curves of other relative indices have only one intersection within the confidence line. Frost days (FD) has a crossing point in the confidence interval, and there was an abrupt change in 1988. In addition, there is an intersection of the summer days (SU) in the confidence interval and an abrupt change point occurred in 1998, tropical nights (TR) has an intersection, and there was an abrupt change in 1999.

#### 3.4.3. Extremal Indices

In this study, the M–K test analysis was carried out on the five extremal indices in the Yangtze River Basin (Figure 10). The positive and negative serial curves of the diurnal temperature range (DTR) have several intersection points within the confidence line. The sliding *t*-test was also performed and its intersections are not abrupt change points; the positive and negative serial curves of other relative indices have only one intersection point within the confidence line. The minimum value of daily maximum temperature (TXn) has a crossing point in the confidence interval, and there was an abrupt change in 1985. The minimum value of daily minimum temperature (TNn) has a crossing point in the confidence interval and the abrupt change occurred in 1981, the maximum value of daily maximum temperature (TXx) has an intersection, and there was an abrupt change in 2001, and the maximum value of daily minimum temperature (TNx) has an intersection in the confidence interval, and an abrupt change in 1998.

### 3.5. The Prediction of Extreme Temperature Indices

From 1970 to 2014, the stability of extreme temperature indices was significantly various (Figure 11). It can be seen from Figure 11 that SU and TR respectively have absolute indices for extreme temperature, and the distribution of each indicator data is relatively concentrated. The median and average values are greater than 0, the trend coefficient is positive, and the trend will continue to maintain the present trend.

TNx, TXx, TNn, and TXn represent the extreme indices of extreme temperature, and the data distribution is relatively concentrated and relatively stable (Figure 12). The extreme indices of FD, TX10, and TN10 median and average are less than 0, the trend coefficient is negative and it shows an increasing trend, the data are rather scattered, and the downward tendencies and instability of the four indices are decreasing. The TX90 and TN90 indicate the extreme indices of relative indices, where the median and average values of TX90 and TN90 are greater than 0, the trend coefficient is positive and it shows an increasing trend, the data are rather scattered, and the upward tendencies and instability of the four indices are increasing.

Moreover, the R/S analysis (Figure 13) also shows that FD, TX10, and TN10 will continue to decrease in the future. TN90, TR, TX90, and SU will keep increasing for some years to come (Table 2 and Table 3). As a result, the Yangtze River Basin continues to warm up, and the risk of extreme temperature events in the basin increases significantly.

### 3.6. Possible Causes of Observed Changes in Temperature Extremes

The most areas in the Yangtze River Basin are located in the eastern monsoon region, so the temperatures in these areas are significantly affected by the atmospheric circulation. Studies have shown that Atlantic Multidecadal Oscillation (AMO) makes contributions to the warming of eastern Asia, strengthening the eastern Asian summer monsoon and weakening the eastern Asian winter monsoon, to a certain extent [46]. Through observation, analysis, and multi-mode simulation, it is found that the warm (positive) phase of AMO not only corresponds to the warm winter in most parts of China, but also warms eastern Asia in every season [47]. Niu et al., drew a similar conclusion by analyzing the correlation between the extreme temperature indices and AMO in the Yangtze River Basin [27]. Some scholars also consider that the number of high temperature days is positively correlated with the area and intensity of subtropical anticyclone. When the days with high temperature increase, the area of subtropical anticyclone increases, and the ridge point of subtropical anticyclone extends westward [48,49]. In addition, the activities of tropical cyclones or typhoons tend to weaken the Western Pacific subtropical anticyclone, and the number of high temperature days may be related to the number and influence the degree of typhoons [50].

On the other hand, human activities have greatly increased the risks of weather with extreme temperature indices [51,52]. Urbanization is the result of the continuous increase of population, the rapid development of economy, and the expansion of urban land. It has an impact on urban temperature, humidity, and precipitation. Among the effects, the heat island effect is the most prominent, which is one of the reasons for the rising trend of extreme temperature. The Yangtze River Basin is an important economic belt in China, with a high level of economic development and urbanization, including three major urban agglomerations in the Yangtze River Delta, the middle reaches of the Yangtze River, and the Mintuo River basin District. The Defense Meteorological Satellite Program (DMSP) has the operational line scanner (OLS), which provides a new data source for collecting data of the dynamic urban expansion on a large spatial scale [53]. The urban night lighting figure can reflect the dynamic expansion information, urbanization, its related land use change, and high-energy consumption. The following figures are the night lighting figures of the Yangtze River Basin in 1992 and 2013, respectively (Figure 14).

Among them, the DN (digital number) values of the night lighting intensity range from 0 to 63, the pixel DN value of the green area is 0, and the green area is the background area, indicating that there is no lighting; when the pixel DN value of the red area is greater than 0, it is the lighting area, and the color shades indicate the lighting intensity. In 1992, the city size in the Yangtze River Basin was relatively small and mostly concentrated in the core urban agglomerations, namely the urban agglomeration in the middle reaches of the Yangtze River, in the Mintuo River Basin, and in the Yangtze River Delta. In 2013, there were great changes. On the basis of the continuous expansion of the surrounding areas of the three major urban agglomerations, more remote areas had the observable lit pixels, and the urbanization level of the whole basin had been significantly improved. The main extreme temperature indices changed significantly in 2013. Except for TNn and TX10, the main extreme temperature indices, such as SU, TR, TNx, TXx, TXn, DTR, TX90, TN90, TN10, and TN90, showed an obvious upward or downward trend. FD and ID also maintained a downward trend (Table 4). This indicates that the urban heat island effect is one of the important reasons for the occurrence of extreme temperature in the Yangtze River Basin.

## 4. Discussions

### 4.1. Comparison with Previous Studies

This paper mainly discusses the variation patterns of extreme temperature in the Yangtze River Basin from 1970 to 2014. In terms of temporal variation pattern, the main warm indices of meteorological stations in the Yangtze River Basin showed an upward trend, while the cold indices showed a downward trend, which is not only consistent with the previous research results in the same region, but is also in good agreement with that reported in Loess Plateau, China, Central Asia, Europe, and globally [47]. Spatially, the main warm indices also presented an increasing tendency, while the cold indices a decreasing tendency, which is consistent with the previous research results in the same region, and also in accordance with many studies in other regions [54]. The main cold indices changed abruptly in the 1980s and the main warm indices changed abruptly in the late 1990s and early 2000s, which is similar to the results of other scholars.

### 4.2. The Effects of Extreme Temperature Indices

Comparing with other regions, for the Plateau areas and the Taihu Lake Basin in the Yangtze River Basin, both the increasing trend shown by the main warming indices and the decreasing tendency presented by for the main cooling indices have remarkable changes. Extreme temperatures at high altitudes are more responsive to global warming [55,56]. In particular, the decreasing number of frost days and ice days is much more beneficial to the plateau grass turning green and livestock wintering. China’s economy has been developing rapidly and the urbanization rate has been increasing since the reform development around 1980, which could be one reason why the extreme temperature indices began to be abrupt around the 1980s. Extreme temperatures affect human health. The frequent heat waves not only threaten lives and but also increase the risks of related diseases and lead to more and more people deceased every year [57]. In addition, they can affect agricultural development. The average annual yield loss of rice in the Yangtze River Basin increased significantly from 8.9% in the 1970s to 17.9% in the early 21st century due to the high temperatures [58,59]. Although the number of frost days and ice days in the river basin has decreased, the Yangtze River Basin, mainly located in the monsoon climate region with a large temperature variation, is prone to spring cold, and a large variation in the number of frost days and freezing days (Figure 4), which increases the affected area of crops [60,61]. In addition, the increase of extreme temperature is helpful to the winter of diseases and pests [62,63], which threaten agricultural production. Extreme temperatures will continue to occur in the future (Table 3). Therefore, extreme temperature prediction and monitoring must be strengthened. Promoting the sustainable development of agriculture, society, and economy in the Yangtze River Basin is important. This paper systematically and comprehensively expounds on the temporal and spatial variation trend of extreme temperature, the prediction of extreme temperature, and the causes and effects of extreme temperature in the Yangtze River Basin. The above methods have certain reference significance for the study of extreme temperature in economically developed and densely urban areas.

## 5. Conclusions

The Yangtze River Basin is an important economic and urban belt in China and the engine of China’s social and economic development. The analysis of the extreme temperature is important for a high impact sustainable development of social economy in the Yangtze River Basin. The study of the characteristics of extreme temperature in this area could help governments and decision makers to make better informed decisions regarding urban construction planning and economic development planning. This paper assesses the temporal and spatial variation analysis of extreme temperature indices of the Yangtze River Basin. The main conclusions are as follows:(1)The trend of cold days, cold nights, ice days, and frost days decreased by −2.2, −3.6, −0.66, and −2.5 d/10a, respectively, while the trend of TX90, TN90, SU, TXx, and TR shows trends of 4.73, 3.82, 2.2, 0.27, and 2.8 d/10a, respectively. The tendency rates of TXn, TNn, TNx, and DTR range is 0.39, 0.5, 0.24, and −0.003 °C/10a, respectively. Spatially, the main extremely warm indices of meteorological stations were increasing, while the extremely cold indices were decreasing in the Yangtze River Basin.(2)Except for DTR and TN90, there were no abrupt changes; the other 11 extreme temperature indicators all had abrupt changes. TX10 changed abruptly in 1987 and TN10 changed abruptly in 2003; ID changed abruptly in 1982 and FD changed abruptly in 1988; SU had an abrupt change point in 1988 and TR had an abrupt change point in 1985; the occurrences of abrupt changes of TXn and TNx were in 2001 and 1998, respectively. The main cold indices changed abruptly in the 1980s and the main warm indices changed abruptly in the late 1990s and early 2000s.(3)The extreme temperature indices are affected by the atmospheric circulation and urban heat island effect in the Yangtze River Basin. Relative indices and absolute indices will continue to maintain the present trend in the future, which has a certain guiding significance for agricultural and social economic development.

In conclusion, the main cold indices of extreme temperature indices showed a decreasing trend, the main warm indices of extreme temperature indices showed an increasing trend, in the Yangtze River Basin, and cold indices and warm indices will continue to maintain the present trend in the future. Therefore, extreme temperature prediction and monitoring must be strengthened to reduce losses caused by extreme temperature disasters, and to promote the sustainable development in Yangtze River Basin.

## Figures and Tables

**Figure 1 ijerph-18-10936-f001:**
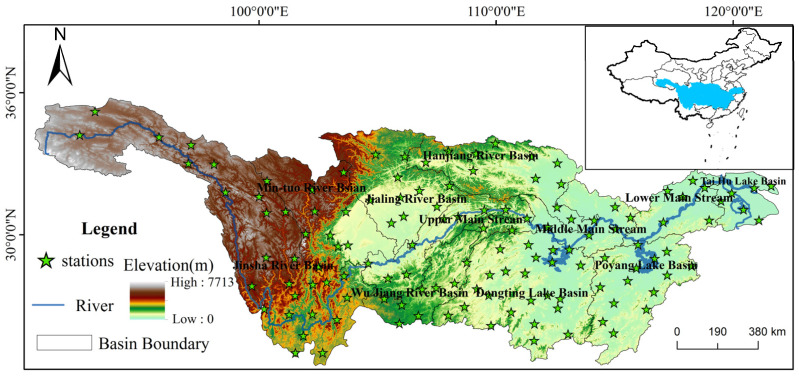
Location, basin boundaries, altitudinal variation range, and distribution of the meteorological stations in the Yangtze River Basin, China.

**Figure 2 ijerph-18-10936-f002:**
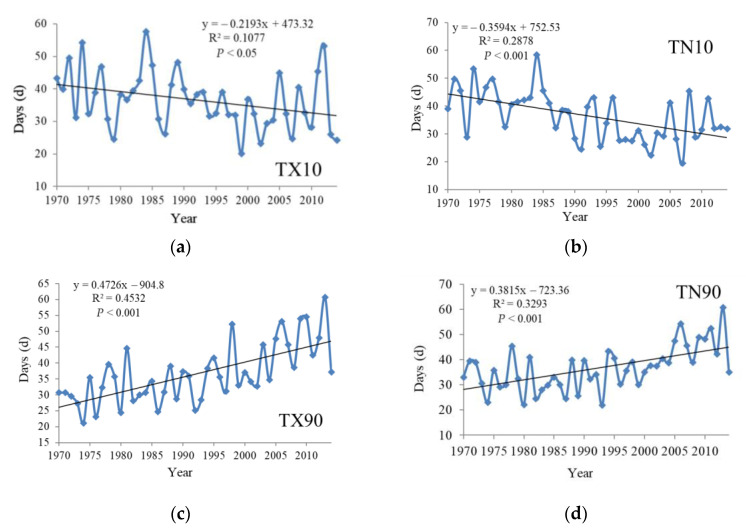
Temporal variation of extreme temperature relative indices in the Yangtze River Basin from 1970 to 2014. (**a**) is TX10, (**b**) is TN10, (**c**) is TX90 and (**d**) is TN90.

**Figure 3 ijerph-18-10936-f003:**
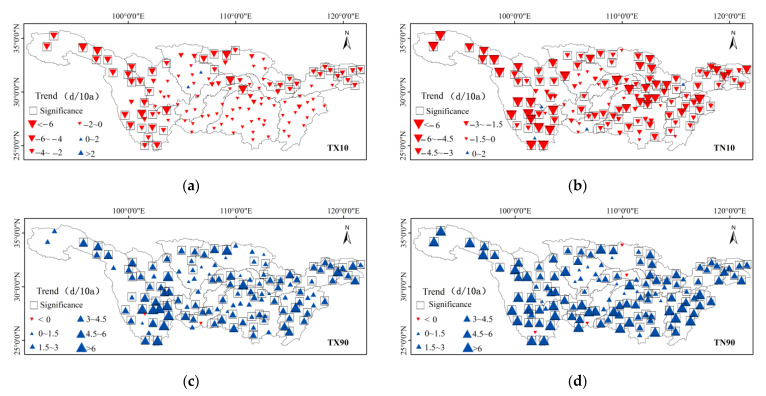
Spatial distribution of interannual variability of relative indices of extreme temperature in the Yangtze River Basin from 1970 to 2014. (**a**) is TX10, (**b**) is TN10, (**c**) is TX90 and (**d**) is TN90.

**Figure 4 ijerph-18-10936-f004:**
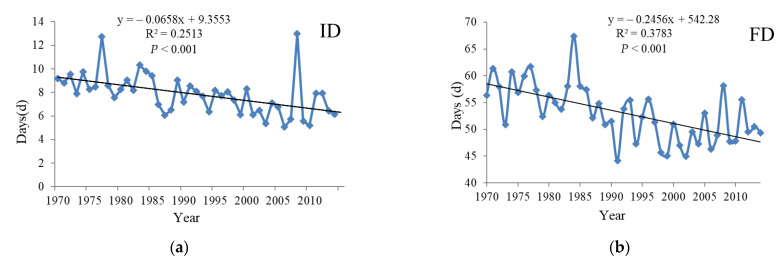
Temporal variation of the absolute indices of extreme temperature in the Yangtze River Basin from 1970 to 2014. (**a**) is ID, (**b**) is FD, (**c**) is SU and (**d**) is TR.

**Figure 5 ijerph-18-10936-f005:**
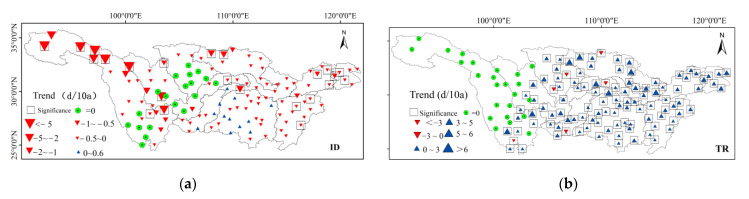
Spatial distribution of annual variation of absolute indices of extreme temperatures in the Yangtze River Basin from 1970 to 2014. (**a**) is ID, (**b**) is TR, (**c**) is SU and (**d**) is FD.

**Figure 6 ijerph-18-10936-f006:**
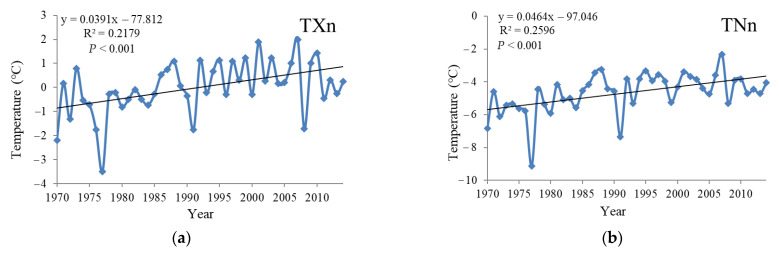
Temporal variation of extreme indices of the extreme temperatures in the Yangtze River Basin from 1970 to 2014. (**a**) is TXn, (**b**) is TNn, (**c**) is TXx, (**d**) is TNx and (**e**) is DTR.

**Figure 7 ijerph-18-10936-f007:**
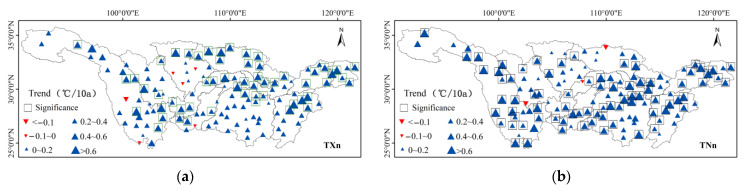
Spatial distribution of extreme indices of extreme temperatures in the Yangtze River Basin from 1970 to 2014. (**a**) is TXn, (**b**) is TNn, (**c**) is TXx, (**d**) is TNx and (**e**) is DTR.

**Figure 8 ijerph-18-10936-f008:**
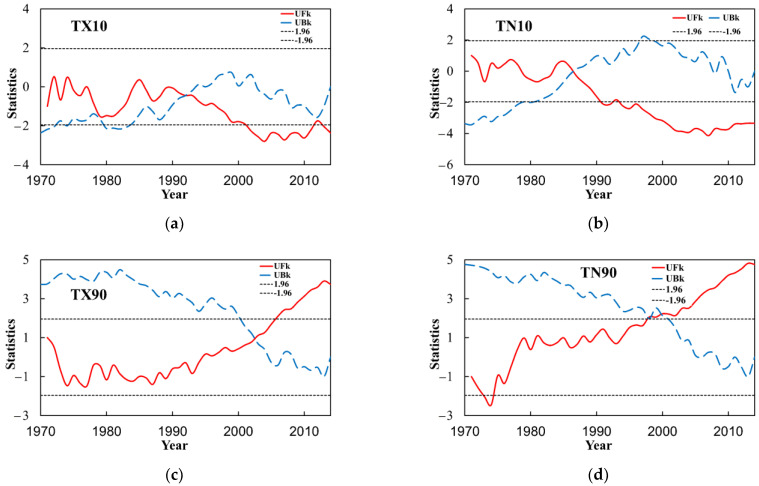
M–K abrupt change test of the relative temperature indices of the Yangtze River Basin from 1970 to 2014. (**a**) is TX10, (**b**) is TN10, (**c**) is TX90 and (**d**) is TN90.

**Figure 9 ijerph-18-10936-f009:**
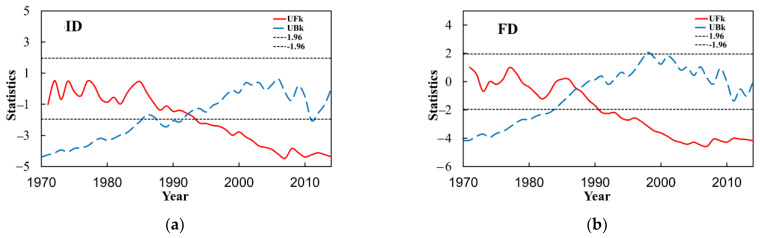
M–K abrupt change test of the absolute indices of extreme temperatures in the Yangtze River Basin from 1970 to 2014. (**a**) is ID, (**b**) is FD, (**c**) is SU and (**d**) is TR.

**Figure 10 ijerph-18-10936-f010:**
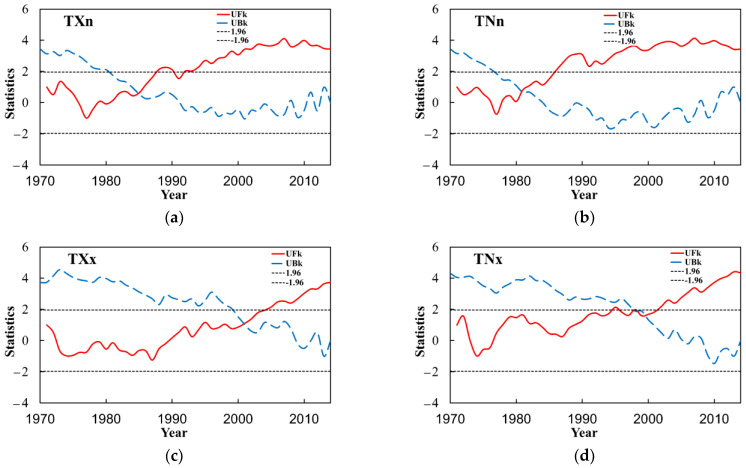
M–K abrupt change test of the extremal indices of extreme temperatures in the Yangtze River Basin from 1970 to 2014. (**a**) is TXn, (**b**) is TNn, (**c**) is TXx, (**d**) is TNx and (**e**) is DTR.

**Figure 11 ijerph-18-10936-f011:**
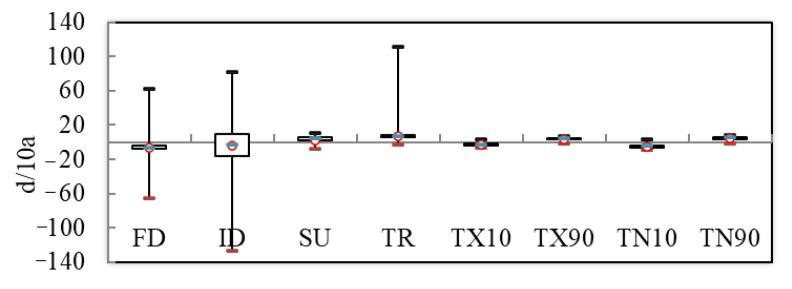
Box-plot of relative indices and absolute indices. At the top of the box-framed figure is the lower quartile value of the sequence, and at the bottom is the upper quartile value. Unit: FD, ID, SU, TR, TX10, RR1, TX90, TN10, TN90 (d/10a).

**Figure 12 ijerph-18-10936-f012:**
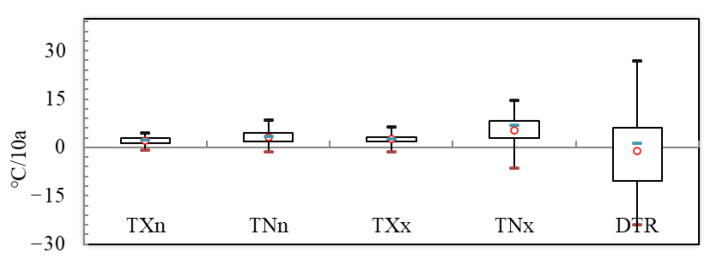
Box-plot of extremal indices. At the top of the box-framed figure is the lower quartile value of the sequence, and at the bottom is the upper quartile value. Unit: TXn, TNn, TXx, TNX, DTR (°C/10a).

**Figure 13 ijerph-18-10936-f013:**
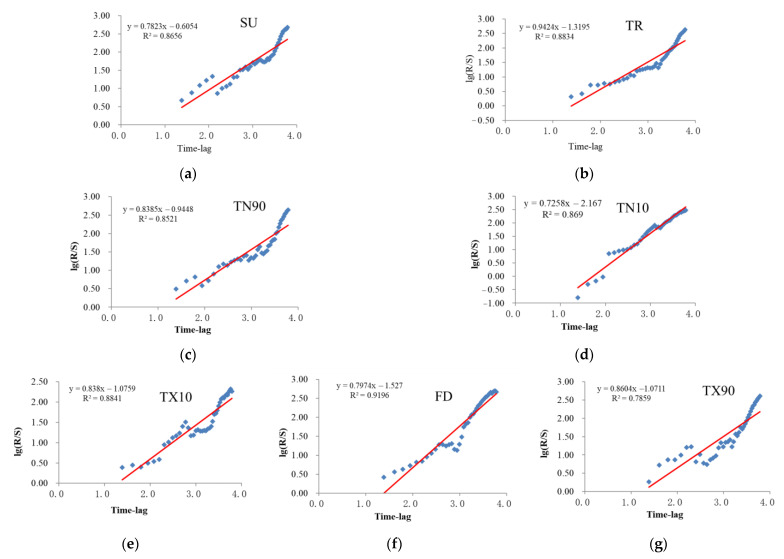
The extreme temperature indices of R/S analysis results in Yangtze River Basin from 1970 to 2014. (**a**) is SU, (**b**) is TR, (**c**) is TN90, (**d**) is TN10, (**e**) is TX10, (**f**) is FD and (**g**) is TX90.

**Figure 14 ijerph-18-10936-f014:**
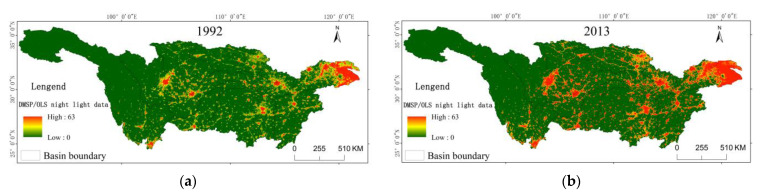
Distribution of DMSP/OLS light data in the Yangtze River Basin. (**a**) is the DMSP/OLS light data in 1992; (**b**) is the DMSP/OLS light data in 2013.

**Table 1 ijerph-18-10936-t001:** 13 Definitions of extreme temperature indices.

Category (1) ^1^	Indices	Descriptive Name	Definitions	Units
Relative indices	TX10	Cold days	Number of days with T_max_ < 10th percentile	days
	TN10	Cold nights	Number of days with T_min_ < 10th percentile	days
	TX90	Warm days	Number of days with T_max_ > 90th percentile	days
	TN90	Warm nights	Number of days with T_min_ > 90th percentile	days
Absolute indices	ID	Ice days	Annual count when TX (daily minimum temperature) < 0 °C	days
	FD	Frost days	Annual count when TN (daily maximum temperature) < 0 °C	days
	SU	Summer days	Annual count when TX > 25 °C	days
	TR	Tropical nights	Annual count when TN > 20 °C	days
Extremal indices	TXn	Coldest day	Annual lowest TX	°C
	TNn	Coldest night	Annual lowest TN	°C
	TXx	Warmest day	Annual highest TX	°C
	TNx	Warmest night	Annual highest TN	°C
	DTR	Diurnal temperature range	Monthly mean difference between TX and TN	°C

^1^ Category (2) Extremely cold temperature indices: TX10, TN10, ID, FD, TXn, and TNn. Extremely warm temperature indices: TX90, TN90, SU, TR, TXx, and TNx.

**Table 2 ijerph-18-10936-t002:** Results of R/S analysis of extreme temperature in indices.

Extreme Temperature Indices	H	R^2^
FD	0.7974	0.9196
SU	0.7823	0.8656
TR	0.9424	0.8834
TX10	0.8380	0.8841
TX90	0.8604	0.7859
TN90	0.8385	0.8521
TN10	0.7258	0.8690

**Table 3 ijerph-18-10936-t003:** The forecast of trends of extreme temperature indices.

Extreme Temperature Indices	Historical Change Tendency	H	Future Change Tendency
FD	decrease	0.7974	decrease
SU	increase	0.7823	increase
TR	increase	0.9424	increase
TX10	decrease	0.8380	decrease
TX90	increase	0.8604	increase
TN90	increase	0.8385	increase
TN10	decrease	0.7258	decrease

**Table 4 ijerph-18-10936-t004:** The trend change of 13 extreme temperature indices was affected by urbanization in two different periods.

Indices	1970–1992	1993–2014
FD	−3.13 d/10a	−0.098 d/10a
ID	−0.729 d/10a	−0.27 d/10a
SU	−1 d/10a	2.74 d/10a
TR	1 d/10a	0.4 d/10a
TN_X_	0.62 °C/10a	−0.16 °C/10a
TX_X_	0.092 °C/10a	0.62 °C/10a
TXn	0.19 °C/10a	0.443 °C/10a
TNn	0.789 °C/10a	−0.092 °C/10a
DTR	−0.129 °C/10a	0.066 °C/10a
TX10	−0.989 d/10a	0.865 d/10a
TX90	−0.788 d/10a	8.479 d/10a
TN10	−4.51 d/10a	0.375 d/10a
TN90	1.49 d/10a	7.837 d/10a

## Data Availability

The data presented in this study are available on request from the corresponding author.

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
