# Peer review of "Assessment on Temporal and Spatial Variation Analysis of Extreme Temperature Indices: A Case Study of the Yangtze River Basin"

_ijerph, 2021, doi:10.3390/ijerph182010936_

Round 1

Reviewer 1 Report

Thank you for your manuscript. Overall I felt the paper was well presented and would be of interest to readers with minor changes. The abstract could be improved with a single sentence summarizing your results to a broad audience. The methods should expand on the rationale of study site choice and the discussion should emphasize the utility of the study beyond the study site for a global audience. Given the journal's focus on public health, I suggest incorporating a section into the introduction connecting extreme temperatures to public health challenges. This will also help connect to your discussion.

Author Response

Response to Reviewer 1 Comments

Thank you for your manuscript. Overall I felt the paper was well presented and would be of interest to readers with minor changes.

Point 1. The abstract could be improved with a single sentence summarizing your results to a broad audience.

Response 1: We thank the reviewer for the valuable comment. We added this point in revised manuscript and the detailed revision can be found in Line 36-39, Page 1.

In short, the main cold indices of extreme temperature indices showed a decreasing trend, the main warm indices of extreme temperature indices showed an increasing trend, and cold indices and warm indices will continue to maintain the present trend in the future, in the Yangtze River Basin.

Point 2. The methods should expand on the rationale of study site choice and the discussion should emphasize the utility of the study beyond the study site for a global audience.

Response 2: (1) We thank the reviewer for the valuable comment. As for the rationale of study site choice, we added this point in revised manuscript and the detailed revision can be found in Line 156-158, Page 4 and Line 161-162, Page 4

The Yangtze River Basin is also a densely populated area in China. Strengthening the research on extreme temperature is conducive to reducing the impact of extreme temperature on public health problems. In short, studying extreme temperature in the Yangtze River Basin is conducive to ensure the sustainable development of society and economy.

(2) As for the utility of the study in discussion, we added this point in revised manuscript and the detailed revision can be found in Line 604-609, Page 20.

This paper systematically and comprehensively expounds on the temporal and spatial variation trend of extreme temperature, the prediction of extreme temperature, the causes and effects of extreme temperature, in the Yangtze River Basin. The above methods have certain reference significance for the study of extreme temperature in economically developed and densely urban areas.

Point 3. Given the journal's focus on public health, I suggest incorporating a section into the introduction connecting extreme temperatures to public health challenges. This will also help connect to your discussion.

Response 3: We thank the reviewer for the valuable comment. We added this point in revised manuscript and the detailed revision can be found in Line 117-129, Page 3.

Another reason to study extreme temperatures in the Yangtze Basin is that the frequent occurrence of heat waves has become an important public health problem [31]. The IPCC assessment report points out that since the mid-20th century, due to global warming and urban heat island effect, summer heat waves have occurred frequently in the world [1]. Studies show that heat waves are linked to heatstroke and heat-related illnesses, leading to an increase in deaths among residents. There are many studies on the health effects of high temperature heat waves in Europe and North America. During the heat wave, a large number of people died directly or indirectly. These deaths are mainly concentrated in the elderly and in persons with some underlying diseases [32-34]. In China, studies on the health effects of heat waves have focused on large cities such as Wuhan City, Nanjing City and Shanghai City in the Yangtze River basin [35-37]. The Yangtze River basin is an important economic zone and urban zone in China. The study of extreme temperature can help to prevent and monitor the impact of heat wave on public health.

Reviewer 2 Report

The paper provides important information about the increasing and decreasing trend of extreme event indices in the Yangtze River Basin. This study is exhaustive and involves a large amount of data from meteorological stations. The authors developed the study according to the proposed objectives and properly analyzed the results using an established methodology.

In order to contribute to the paper, I suggest that the authors make the following adjustments:

- Reverse the order of the Study Area and Data used;

- Include the location of the study area in China in Figure 1 so that the reader who does not know about the study area can identify it;

- Describe what the acronym DMSP/OLS is;

- Move the results described in item 4 Discussion to item 3 Results.

- Add a general conclusion to the paper.

Author Response

Response to Reviewer 2 Comments

The paper provides important information about the increasing and decreasing trend of extreme event indices in the Yangtze River Basin. This study is exhaustive and involves a large amount of data from meteorological stations. The authors developed the study according to the proposed objectives and properly analyzed the results using an established methodology.

In order to contribute to the paper, I suggest that the authors make the following adjustments:

Point 1. Reverse the order of the Study Area and Data used;

Response 1: Thank you for pointing this out. We have reversed the Study Area and Data and the detailed revision can be found in Line 142-150, Page 3, Line 151-165, Page 4 and Line 188-209, Page 5.

Point 2. Include the location of the study area in China in Figure 1 so that the reader who does not know about the study area can identify it;

Response2: We thank the reviewer for the valuable comment. We added this point in revised manuscript and the detailed revision can be found in Line 162-163, Page 4.

Point 3. Describe what the acronym DMSP/OLS is;

Response 3: We thank the reviewer for the valuable comment. The description of DMSP/OLS is DMSP (Defense Meteorological Satellite Program) / OLS (Operational Linescan System) night light data.

We added this point in revised manuscript and the detailed revision can be found in Line 198-199, Page 5.

Point 4. Move the results described in item 4 Discussion to item 3 Results.

Response 4: We agree with the reviewer's comment. we have moved the results described in item 4 Discussion to item 3 Results and the detailed revision can be found in Line 468-507, Page17 and Line 508-517, Page18.

Point 5. Add a general conclusion to the paper.

Response 5: We thank the reviewer for the valuable comment. We added this point in revised manuscript and the detailed revision can be found in Line 611-627, Page 20 and628-644, Page 21.

The Yangtze River Basin is an important economic and urban belt in China and the engine of China's social and economic development. The analysis of the extreme temperature is important for its high impact Sustainable development of social economy in the in the Yangtze River Basin. This study of the characteristics of extreme temperature in this area would help governments and decision makers to make better informed decisions regarding urban construction planning and economic development planning. This paper assessment on temporal and spatial variation analysis of extreme temperature indices of the Yangtze River Basin. The main conclusions are as follows:

(1) The trend of cold days, cold nights, ice days, frost days decreased by -2.2, -3.6, -0.66, -2.5 d/10a, respectively, while the trend of TX90, TN90, SU, TXx, TR shows trends of 4.73, 3.82, 2.2, 0.27, 2.8 d/10a. The tendency rates of TXn, TNn, TNx, DTR range is 0.39, 0.5, 0.24, -0.003 °C/10a. Spatially, the main extremely warm indices of meteorological stations were increasing, while the extremely cold indices were decreasing, in the Yangtze River Basin.

(2) Except for DTR and TN90 were no abrupt changes, the other 11 extreme temperature indicators all had abrupt changes. TX10 changed abruptly in 1987 and TN10 changed abruptly in 2003; ID changed abruptly in 1982 and FD changed abruptly in 1988; SU had an abrupt change point in 1988 and TR had an abrupt change point in 1985; the occurrences of abrupt changes of TXn and TNx were in 2001 and 1998, respectively. The main cold indices changed abruptly in the 1980s and the main warm indices changed abruptly in the late 1990s and early 2000s.

(3) The extreme temperature indices are affected by the atmospheric circulation and urban heat island effect, in the Yangtze River Basin. Relative indices and absolute indices will continue to maintain the present trend in the future, which has certain guiding significance for agricultural and social economic development.

In conclusion, the main cold indices of extreme temperature indices showed a decreasing trend, the main warm indices of extreme temperature indices showed an increasing trend, in the Yangtze River Basin, and cold indices and warm indices will continue to maintain the present trend in the future. Therefore, extreme temperature prediction and monitoring must be strengthened, and to reduce losses caused by extreme temperature disasters, and to promote the sustainable development in Yangtze River Basin.
